# Post-COVID-19 Syndrome in Neurology Patients: A Single Center Experience

**DOI:** 10.3390/pathogens12060796

**Published:** 2023-06-02

**Authors:** Elena Hegna, Valentino Rački, Mario Hero, Eliša Papić, Gloria Rožmarić, Klara Radović, Vita Komen, Marina Bralić, Marina Legac Škifić, David Bonifačić, Zoran Tomić, Olivio Perković, Vladimira Vuletić

**Affiliations:** 1Department of Neurology, General Hospital Pula, 52100 Pula, Croatia; 2Department of Neurology, Clinical Hospital Center Rijeka, 51000 Rijeka, Croatiamarinals@student.uniri.hr (M.L.Š.); zorantomic554@gmail.com (Z.T.);; 3Department of Neurology, Faculty of Medicine, University of Rijeka, 51000 Rijeka, Croatia; 4Department of Emergency Medicine, University Hospital Center Rijeka, 51000 Rijeka, Croatia

**Keywords:** COVID-19, post-COVID-19 syndrome, headache, cognitive dysfunction

## Abstract

Our aim was to determine the frequency and characteristics of neurological post-COVID-19 syndrome and the diagnostic and therapeutic measures that were used for the treatment of these patients. Data were collected for 243 patients examined during the period of 11 May 2021 to 22 June 2022. The inclusion criteria were COVID-19 illness and neurological symptoms associated with COVID-19. The exclusion criteria were non-neurological symptoms, patients who did not suffer from COVID-19, and symptoms that occurred after vaccination against the SARS-CoV-2 virus. Data for 227 patients with neurological post-COVID-19 symptoms were analyzed. Most patients presented with multiple symptoms, most often headache, cognitive impairment, loss of smell, paresthesia, fatigue, dizziness, and insomnia. Patients were most often referred for consultative examinations, neuroradiological imaging, and EEG. The therapy was mostly symptomatic. Most patients had no change in their symptoms on follow-up visits (53.21%), while positive outcome was found in 44.95% of patients. This study found that neurological post-COVID-19 syndrome appears to be more common in women, and generally, the most common symptoms are headache and cognitive impairment. The gender distribution of symptoms was clearly visible and should be further investigated. There is a need for longitudinal follow-up studies to better understand the disease dynamic.

## 1. Introduction

Coronavirus disease (COVID-19) is an infectious disease caused by the SARS-CoV-2 virus [1]. In Croatia, approximately 1.27 million cases of COVID-19 infection and 18,230 deaths caused by COVID-19 were reported. To this day, approximately 5.36 million doses of the vaccine have been administered, with 2.32 million people having received at least one dose of the vaccine (57.60% of the population) and 2.25 million people who were fully vaccinated (55.87% of the population). The vaccination rate in Croatia is below average, considering that the average percentage of the fully vaccinated population in Europe is 66.24% [2]. More strikingly, the death per capita rates put Croatia in seventh place in the world, with approximately 4523 deaths per 1 million inhabitants, highlighting the great toll that the pandemic had on society [3]. The burden of COVID-19 is not measured only in deaths, and unfortunately, a large proportion of patients have continuous post-COVID-19 problems [4].

Post-COVID-19 syndrome is a multisystemic disorder that includes about 50 symptoms [5]. Due to the lack of a definition for post-COVID-19 syndrome, the WHO issued the Delphi guidelines, which define post-COVID-19 syndrome as a condition that occurs in individuals with a history of SARS-CoV-2 infection, usually 3 months after the onset of infection, with symptoms that last at least 2 months and cannot be explained by an alternative diagnosis. Common symptoms include fatigue, shortness of breath, and cognitive dysfunction. Symptoms may persist from the initial infection or may arise de novo after recovery from the disease [6]. Although COVID-19 is primarily presented with respiratory symptoms, neurological manifestations of this disease are being increasingly recognized [7].

There are two ways for the entry of theSARS-CoV-2 virus into the central nervous system (CNS). The first viable way of invasion into the CNS is through the olfactory nerve. It is thought that SARS-CoV-2 can enter the nervous system by crossing the neural-mucosal barrier in the olfactory mucosa and penetrating the neuroanatomical areas, including the primary respiratory and cardiovascular centers in the medulla oblongata [8,9]. There is clear evidence that SARS-CoV-2 can cause significant morphological and functional changes in the CNS, which could be the basis for the development of long-term post-COVID-19 syndromes [10,11].

It has been suggested that the chronic neurological symptoms of post-COVID-19 syndrome be classified into four groups: (I) cognition, mood, and sleep disorders; (II) dysautonomia; (III) pain syndromes; and (IV) exercise intolerance [12]. The prevalence of the most common neurological COVID-19 symptoms included fatigue (32%), myalgia (20%), taste impairment (21%), smell impairment (19%), and headache (13%) [13]. Brain fog and depression/anxiety also occur frequently during follow-up, suggesting their relevance to long-term COVID syndrome [14].

Studies have shown that neurological post-COVID-19 symptoms can persist for even a year after the initial infection. When symptoms are compared between the 3-month follow-up and 1-year follow-up examination, there is an improvement in patient outcomes. However, a portion of patients experience new-onset neurological symptoms [15]. There is still not enough data on the outcomes of the neurological post-COVID-19 syndrome.

Due to the significant burden of COVID-19 in Croatia and large patient numbers during previous pandemic waves, we have established a separate post-COVID-19 outpatient clinic to facilitate faster examinations and follow-ups. The large number of patients and the variety of symptoms led to the aim of this study, which was to retrospectively assess the frequency and characteristics of neurological symptoms of post-COVID-19 syndrome in our outpatient clinic, including the diagnostic and therapeutic measures that were taken in the treatment of these patients and what outcomes the patients had.

## 2. Materials and Methods

In this research, we retrospectively collected data for 243 patients examined in the neurological post-COVID-19 outpatient clinic during the period of 11 May 2021 to 22 June 2022 using the in-hospital medical records system. To be included in the research, patients had to meet several inclusion criteria, including having recovered from COVID-19 disease, proven by a positive rapid antigen test performed by a medical professional or a positive PCR test for SARS-CoV-2. The second criterion was having new neurological symptoms persisting after the acute COVID-19 disease or symptoms occurring after the acute infection has subsided. The exclusion criteria were symptoms that were not clearly neurological, patients who did not have COVID-19, and symptoms that only occurred after vaccination with one of the SARS-CoV-2 vaccines. It is important to emphasize that only the symptoms that appeared after the COVID-19 infection were taken into account. Symptoms that were present before the infection were excluded.

The collected data included age, sex, description of symptoms, the time elapsed from infection to the onset of post-COVID-19 symptoms, whether hospitalization was required due to the infection, and the recommended diagnostic and therapeutic measures. The outcome was assessed for all patients via control visits. Statistical data were processed in GraphPad Prism using descriptive statistics. The research was approved by the Ethics Committee of the Clinical Hospital Center Rijeka.

## 3. Results

We collected data for 243 patients; eight were excluded based on not having a COVID-19 infection, while another eight were excluded because symptoms appeared after vaccination against the SARS-CoV-2 virus. One patient presented with symptoms of post-COVID-19 syndrome but also with side effects of the vaccine and the symptoms that occurred after vaccination were excluded. Finally, 227 patients met the inclusion criteria and were therefore included in the data analysis (Figure 1).

The data were analyzed for 227 patients with neurological post-COVID-19 symptoms. Of these, 158 patients (70%) were women, and 69 patients (30%) were men. The mean age of all the patients was 51.89 ± 15.67 years. On average, women were slightly younger than men, with an average age of 50.73 ± 15.17 years. The mean age of men was 54.13 ± 16.58 years. Around 17 patients were previously hospitalized due to a COVID-19 infection, which is 7% of patients, while the other 210 patients recovered from a milder form of the disease (Table 1).

### 3.1. Frequency and Gender Distribution of Post-COVID-19 Symtoms

Most patients presented with multiple symptoms, which resulted in the number of reported symptoms being greater than the number of patients. The most common symptoms were headaches (30%) and cognitive problems (29%). The next most common were changes in the sense of smell (17%), paresthesia (16%), chronic fatigue (15%), and dizziness associated with nausea and vomiting (15%). Approximately 11% of patients presented with insomnia, while 10% of patients complained of myalgias and arthralgias associated with loss of strength. Mood disorders, including depression, anxiety, and mood swings, were present in 9% of patients. Changes in the sense of taste and neuropathic pain were present in the same number of patients (7%). Ataxia, balance disorders, as well as neuropathy, radiculopathy, and autonomic dysfunction were present in 5% of patients. Syncope, presyncope, and collapse due to the similarity of symptoms were grouped into one group and were present in 10 patients, which is approximately 4% of the patients. Visual and auditory symptoms (including tinnitus and impaired hearing) were also present in approximately 4% of patients. Other symptoms (including tremors, hypoesthesia, myoclonus, and partial epilepsy) were present in 5% of the patients (Table 2). The average time between the onset of infection and the onset of symptoms of post-COVID-19 syndrome was 6.11 ± 11.15 weeks.

When the symptoms are distributed by gender, it is clearly visible that the number of almost all symptoms is significantly higher in women. The only exception is the categories of syncope, presyncope, collapse and balance disorders, and ataxia, where the number of reported symptoms was equal in both sexes. The most frequently reported symptom in women was a headache, while men most often complained of cognitive problems. Such results can be explained simply by the fact that women make up slightly more than two-thirds of the patients included in this analysis. Due to the uneven distribution of patients by gender, a much better insight into the differences in the frequency of symptoms gives us the relative proportion of symptoms between genders (Table 2).

While observing the relative shares, it can be noted that there are significant differences in the frequency of symptoms between the sexes. Men more often complained about cognitive impairments (reported by 33% of men and 27% of women), neuropathic pain (10% of men, 6% of women), ataxia and balance disorders (9% of men, 4% of women), syncope, pre-syncope, and collapse (7% men, 3% women), and the category of other symptoms (7% men and 4% women). Headaches occurred more often in women (recorded in 34% of patients) along with smell disorders (18% of women, 14% of men), dizziness accompanied by nausea and vomiting (16% of women, 13% of men), psychiatric disorders (12% of women, 3% of men), insomnia (12% women, 9% men), taste disorders (9% women, 3% men), as well as autonomic (6% women, 3% men) and visual symptoms (4% women, 1% men). Some of the symptoms occurred with equal frequency in both sexes. Such symptoms are paresthesia (16% in both sexes), chronic fatigue and general weakness (15% women, 16% men), myalgias and arthralgias with loss of strength (11% women, 12% men), neuropathy and radiculopathy (5% women, 4% men), and auditory symptoms (3% women, 4% men) (Table 2).

### 3.2. Diagnostic Procedures Used in Assessing the Symptoms

Most patients were referred to multiple diagnostic tests, and therefore, the number of diagnostic procedures is greater than the number of patients. Most often, the patients were sent for consultative examinations to specialists in other branches of medicine, and that represents 22% of all diagnostic procedures and 41% of all patients. Next in order of frequency are the neuroradiological methods to which 37% of patients referred, EEG (31%), and Color Doppler of blood vessels of the head and neck (28%). Around 17% of patients were referred for EMNG and 16% for laboratory blood tests, which included at least one of the tests, such as complete blood count, pituitary hormones, thyroid hormones, and vitamin and mineral concentrations. Additionally, 31 patients (14%) were referred to psychological testing, while 13 patients (6%) were sent for testing with evoked potentials (BAER, VEP, and P300). Meanwhile, 3 patients (1%) were referred for other diagnostic procedures, which included electrophoresis, headache diary, and referral to another neurological outpatient clinic (Table 3).

The most common specialist to whom the patients were referred was an audiologist, to whom 17 patients were referred, followed immediately by a psychiatrist to whom 16 patients were referred. Patients were also sent to an endocrinologist (12 patients), otorhinolaryngologist (10 patients), physiatrist (7 patients), cardiologist (7 patients), psychologist (6 patients), rheumatologist/immunologist (6 patients), ophthalmologist (4 patients), pulmonologist (4 patients), dermatologist/allergist (2 patients), infectious disease specialist (2 patients), and oral pathologist (1 patient) (Table 4).

### 3.3. Therapy Options

Therapy was prescribed to 52 patients, of which 35 patients were prescribed just one drug, 16 patients took two drugs, and one patient was prescribed therapy with three drugs. The most frequently prescribed therapies were vitamin supplements, prescribed to 37 patients (16%); antidepressants and analgesics, which were each prescribed to 7 patients; anxiolytics and hypnotics, which were prescribed to 6 patients; antiepileptics to 5; and physical therapy and histamine analogs were prescribed as therapy to 3 patients. One patient was prescribed melatonin supplements, and one was prescribed beta blockers (Table 5).

### 3.4. Outcomes

Finally, 109 patients had follow-up examinations after the initial assessment in the outpatient clinic, which is 48% of all the patients included in this research. Around 79 of them (72%) had only one follow-up examination, 16 of them had two, 11 had three, and only 3 patients had four follow-up examinations (Figure 2, Table 6).

Patients had mixed outcomes at control visits. The average time from the initial examination to the follow-up was 9.63 ± 5.10 months. About 58 patients (53% of the follow-up patients) had no change in the symptoms observed in the first examination. A total of 49 patients had a positive outcome, which means that the symptoms either completely disappeared or there was a significant improvement. Two patients had negative outcomes with the exacerbation of symptoms. It is worth mentioning that these two patients had COVID-19 again in between the follow-up examinations. If we look at patient outcomes regarding the number of follow-up examinations, we can see that the distribution is even. Patients in whom there was no change in symptoms prevail in all groups, regardless of the number of follow-up examinations. The relative share of patients with a positive outcome compared to the total number of patients in that group is the highest in the group with one follow-up examination (49%). Two patients with negative outcomes were in the groups with two and three follow-up examinations (Table 6).

## 4. Discussion

We present the results from the neurological post-COVID-19 outpatient clinic of CHC Rijeka for the period of 11 May 2021 to 22 June 2022. An analysis of 227 patients revealed that headache (30%) and cognitive impairments (29%) were the most common neurological post-COVID-19 symptoms. Such results are expected, considering that these are one of the most common neurological disorders in the general population. According to some studies, 46% of the population suffers from headaches [16], while cognitive problems predominantly affect the elderly population and can affect up to 70% of the population over 60 years of age [17]. Additionally, a significant portion of patients complained about smell disorders (17%), paresthesia (16%), chronic fatigue (15%), and dizziness, accompanied by nausea and vomiting (15%). According to the meta-analysis by Lopez-Leone and colleagues, the most common neurological symptoms of the post-COVID-19-syndrome are, precisely, chronic fatigue, headache, cognitive complaints, and smell disorders, which is in accordance with the data obtained in our research [18]. Most studies support chronic fatigue as the most common symptom of post-COVID-19 syndrome, which is present in 32% to 58% of post-COVID-19 patients [18,19,20,21], while according to the results of our research, it is the fifth most frequent symptom in the Rijeka Clinical Hospital Center, present in only 15% of patients. Sykes et al. concluded that the symptoms of neurological post-COVID-19 syndrome can be divided into three groups. Group A includes myalgia and chronic fatigue, group B low mood, anxiety and sleep disorders, while group C consists of patients with cognitive impairments [22].

From the data analysis, it is evident that twice as many women were examined in the post-COVID-19 neurological outpatient clinic, which could indicate a higher frequency of post-COVID-19 syndrome in women. A multicenter study by Fernández-de-las-Peñas et al. indicated that the female gender is associated with a higher risk of developing post-COVID-19 syndrome, with an emphasis on depression, anxiety, and sleep disorders [23]. According to the research of Bai et al., women even have a three times greater risk of being diagnosed with post-COVID-19 syndrome [24]. Finally, a recent review by Sylvester et al. highlighted that there are present sex-disaggregated differences for COVID-19 sequelae and long-COVID syndrome, with a higher frequency found in women [25]. The fact that the post-COVID-19 syndrome is more frequent in women may indicate a potential contribution of autoimmunity in the development of this complication of SARS-CoV-2 infection [26]. However, it is important to highlight that the magnitude of the difference could be impacted by the propensity of women to seek healthcare more often, especially in the ages found in our study [27,28]. According to the obtained results, there is a clear difference in the frequency of individual symptoms between genders. For example, in women, headaches were almost twice as common as in men, taste disorders were three times more common, while psychiatric disorders were even four times more common. On the other hand, symptoms such as neuropathic pain, balance disorders, syncope, presyncope, and collapse were two times more common in men. According to the research published so far, there is a difference in the distribution of symptoms between the genders, which does not fully agree with the results obtained in our research. According to previous data, chronic fatigue and psychiatric problems occur more often in women [29,30]. On the other hand, there are studies in which the gender distribution of symptoms is not present, however, they occur in equal proportions in both sexes [31,32]. So far, a small number of studies investigating gender differences in neurological post-COVID-19 syndrome have been conducted, and there is room for research on this topic.

The diagnostic procedures included a very wide range of tests, the most common of which were consultative examinations by specialists from other branches of medicine, which is expected due to the polymorphism of post-COVID-19 syndrome symptoms. Neuroradiological methods, EEG, and neurosonological tests were the next most frequent. Since the most common neurological symptoms of post-COVID-19 syndrome are headaches and cognitive problems, these diagnostic methods are justifiably more frequent. Neuroradiological methods are recommended in all patients with headaches, new neurological deficits, new and sudden severe headaches, HIV-positive patients with a new type of headache, and patients over 50 years old with a new headache [33], and they are undoubtedly indicated in patients with an acute onset of cognitive impairment and/or rapid neurological deterioration [34]. Neuropsychological damage is not always associated with abnormal MRI findings, however, in a subset of patients with post-COVID syndrome, the MRI shows white matter lesions that are not limited to patients with severe disease. Therefore, changes associated with the disease of COVID-19 could be considered in the differential diagnosis of white matter lesions [35,36]. EEG abnormalities are common in patients with COVID-19 and include a wide range of findings, such as background abnormalities, periodic and rhythmic activity, and other epileptiform abnormalities [37]. The surprising result is the relatively low frequency of psychological testing, to which only 14% of patients were referred, as many patients have rejected further diagnostics for the present cognitive dysfunction. Psychological testing is among the main diagnostic tools for the evaluation of cognitive functions [38], and given the relatively high proportion of patients who complained of cognitive impairment, it was expected that this would be one of the most common diagnostic procedures.

Therapy was most often not prescribed and was recommended in 23% of patients, indicating a lack of a clear therapeutic plan in post-COVID-19 patients. Since the etiology of the post-COVID-19 syndrome is still not fully known, therapy is usually limited to symptomatic treatment. Therefore, the prescribed therapy mostly depends on the manifestations of the post-COVID-19 syndrome. The most frequently prescribed therapy was vitamin supplements, followed by antidepressants, analgesics, and anxiolytics. Research by Naureen et al. indicates the possible benefits of using multivitamin dietary supplements in patients suffering from post-COVID-19 syndrome. Vitamin complexes containing B vitamins, vitamin C, vitamin D, acetyl L-carnitine, and hydroxytyrosol could be crucial in the treatment of chronic fatigue [39]. Currently, no pharmaceutical drugs have been shown to alleviate the symptoms of post-COVID-19 syndrome, however, paracetamol and NSAIDs can be used to treat specific symptoms, such as fever and pain [40]. Creating a rehabilitation plan can be helpful for some patients and may include physical and occupational therapy, speech and language therapy, and neurologic rehabilitation of cognitive symptoms. A gradual return to exercise has been hypothesized to have a beneficial effect on post-COVID-19 symptom relief [41]. Early rehabilitation has been shown to be critical for improving patients’ long-term recovery and functional independence, so rehabilitation should begin as soon as possible. The rehabilitation program must be personalized and focused on solving the specific problems of the patient [42].

To the best of our knowledge, only a few prospective studies monitoring the outcome of patients with neurological post-COVID-19 syndrome were published. According to Jennifer A. Frontera et al., patients with neurological complications of COVID-19 had significantly worse outcomes at 6 months [43]. In a 3-month follow-up study conducted in Italy, there was no significant change in the symptoms between the 1-month and 3-month follow-up examinations, which is similar to our results, as no change in the symptoms is the most common outcome [44]. Physical therapy and rehabilitation have been shown to benefit the patient’s recovery and increase the probability of positive functional outcomes [45,46]. It is important to highlight that most patients did not come to follow-up visits, which could indicate a positive outcome and alleviation of symptoms, although this is speculative and cannot be confirmed by this study. There is much to learn about this syndrome, especially relating to the outcomes, and further follow-up is required.

This study has several limitations. First, the time frame for patient assessment was not standardized, so the study included patients with different time intervals since recovering from COVID-19. For this reason, it is not possible to assess whether there is a difference in the frequency of occurrence of post-COVID-19 symptoms regarding the time that has passed since the infection. Secondly, data on pre-existing comorbidities were not collected from the patients. This is very significant because research has shown that people with a greater number of comorbidities have a higher risk of developing post-COVID-19 syndrome [47]. Furthermore, due to the retrospective nature of the study, we were not able to discern which COVID-19 variant was present in our patients and whether this might impact the nature of the symptoms. We have decided not to include patients who had symptoms appear after vaccination with one of the approved vaccines against the SARS-CoV-2 virus, which limits the possibility of assessing the potentiated impact of vaccination on the frequency and intensity of the symptoms.

## 5. Conclusions

We retrospectively analyzed data from the neurological post-COVID-19 outpatient clinic of Rijeka Clinical Hospital Center and identified 227 patients with newly developed neurological symptoms of post-COVID-19 syndrome. The most prominent symptoms were headaches and cognitive problems. In the case of certain symptoms, unequal gender distribution was clearly visible. The most common diagnostic procedures were consultative examinations and neuroradiological methods, which are in accordance with the polymorphism of symptoms and point to the fact that a multidisciplinary approach is essential for the treatment and diagnosis of post-COVID-19 syndrome. Therapy was prescribed only to a small portion of the patients and was mostly symptomatic. The continuation of symptoms was present in most patients coming to follow-up visits, and it would be interesting to follow these patients long-term to see the longevity of this syndrome. Looking at the overall results, there is plenty of room for further research. It is necessary to investigate the differences in the gender distribution of symptoms in more detail, given the fact that very few such studies are available. It would also be interesting to examine the impact of vaccination against COVID-19, as the most effective preventive measure, on the frequency and severity of symptoms related to post-COVID-19 syndrome. Longitudinal follow-up studies of patients are necessary to better understand the dynamics of the onset and disappearance of symptoms and to find a potential causative therapy for the treatment of this syndrome.

## Figures and Tables

**Figure 1 pathogens-12-00796-f001:**
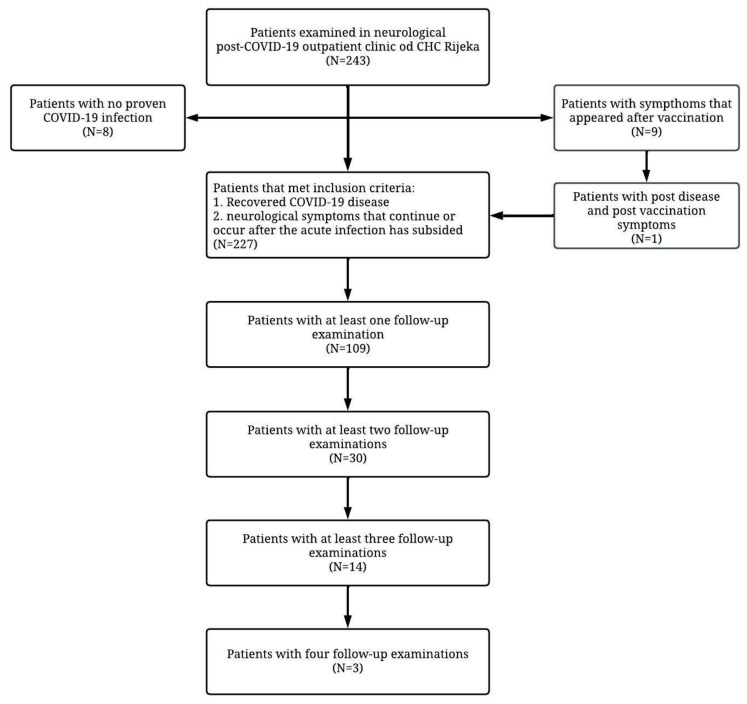
Flowchart of the study indicating included and excluded patients.

**Figure 2 pathogens-12-00796-f002:**
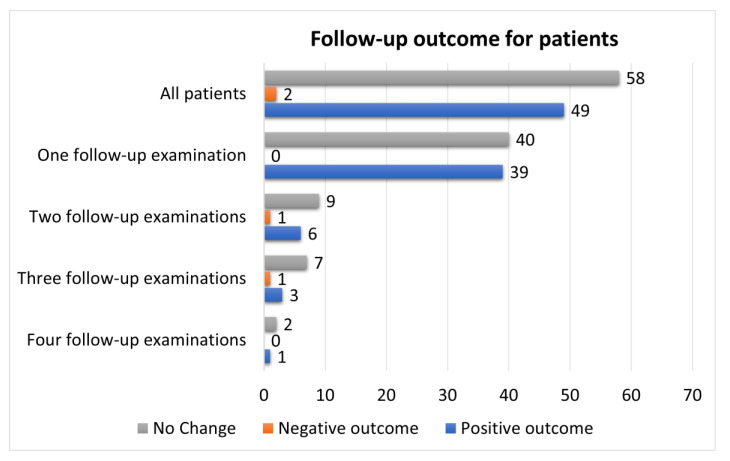
Outcomes of the follow-up examinations. Most of the patients came to follow-up visits, and the symptoms persisted in more than a half of those (*n* = 58, 53%). Overall, looking at most follow-up visits, there was a persistence of post-COVID-19 symptomatology in more than a half, even though the overall number of patients is low in two to four follow-up examinations in our study.

**Table 1 pathogens-12-00796-t001:** Socio-demographic data of the patients included in this study (N = 227).

Parameter	No. or Mean (Standard Deviation)
No. of women	158
No. of men	69
Mean age	51.89 ± 15.67 years
Mean age of women	50.73 ± 15.17 years
Mean age of men	54.13 ± 16.58 years
No. of previously hospitalized patients	17

**Table 2 pathogens-12-00796-t002:** Most frequently reported symptoms in patients.

Reported Symptoms	Total Number of Patients	Number of Male Patients	Number of Female Patients	Relative Frequency of Symptoms in Males/Females (%)
Headache	67	13	54	19%/34%
Cognitive impairment	65	23	42	33%/27%
Smell disorders	39	10	29	14%/18%
Paresthesia	37	11	26	16%/16%
Chronic fatigue	35	11	24	16%/15%
Dizziness associated with nausea and vomiting	34	9	26	13%/16%
Insomnia	25	6	19	9%/12%
Myalgia, arthralgia, loss of strength	23	7	16	12%/11%
Psychiatric disorders	21	2	19	3%/10%
Taste disorders	16	2	14	3%/9%
Neuropathic pain	16	7	9	10%/6%
Ataxia and balance disorders	12	6	6	9%/4%
Neuropathy and radiculopathy	11	3	8	4%/5%
Autonomic dysfunction	11	2	9	3%/6%
Syncope, presyncope and collapse	10	5	5	7%/3%
Auditory symptoms	8	5	3	4%/3%
Visual symptoms	8	7	1	1%/4%
Other symptoms	12	5	7	7%/4%

**Table 3 pathogens-12-00796-t003:** Diagnostic tests recommended to patients.

Diagnostic Tests Recommended to Patients	Number of Patients
Consultative examinations	94
Neuroradiological imaging	83
EEG	70
Neurosonology methods	63
EMNG	39
Laboratory blood tests	37
Psychological testing	313
BAER, VEP and P300	13
Other diagnostic pocedures	3

**Table 4 pathogens-12-00796-t004:** Most frequent consultative examination.

Consultive Examinations	Number of Patients
Audiologist	17
Psychiatrist	16
Endocrinologist	12
Otorhinolaryngology	10
Physiatrist	7
Cardiologist	7
Psychologist	6
Rheumatologist/immunologist	6
Ophthalmologist	4
Pulmonologist	4
Dermatologist/allergologist	2
Infectious disease specialist	2
Oral pathologist	1

**Table 5 pathogens-12-00796-t005:** Therapy prescribed to the patients.

Therapy Prescribed to Patients	Number of Patients
Vitamin supplements	37
Antidepressants	7
Analgetics	7
Anxiolytics and hypnotics	6
Antiepileptics	5
Physical therapy	3
Histamin analogues	3
Melatonin supplements	1
Beta blockers	1

**Table 6 pathogens-12-00796-t006:** Number of patients that had follow-up examinations.

Follow-Up Examinations	Number of Patients	Percentage
One follow-up examination	79	72%
Two follow-up examinations	16	15%
Three follow-up examinations	11	10%
Four follow-up examinations	3	3%

## Data Availability

The data that support the findings of this study are available upon request from the main and corresponding authors. The data are not publicly available due to ethical restrictions.

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
