# Peer review of "Post-COVID-19 Syndrome in Neurology Patients: A Single Center Experience"

_pathogens, 2023, doi:10.3390/pathogens12060796_

Round 1

Reviewer 1 Report

Thank you for trusting me to review this manuscript.

The present study This article summarized and analyzed the frequency and characteristics of neurological post COVID-19 syndrome and the diagnostic and therapeutic measures used in treatment. This study provides reference data for the better understanding of the neurological post COVID-19 syndrome.

The following minor comments are for reference:

1. The major concern is that patients with neurological symptoms preceding infection with COVID-19 have not been excluded. In addition, the authors' conclusion that " This study found that neurological post-COVID-19 syndrome is more common in women" is inappropriate because the number of female patients included in the study was much higher than that of men.

2. In Materials and Methods, authors hold that “To be included in the research, patients had to meet several inclusion criteria, which were a recovered COVID-19 disease proven by a positive rapid antigen test performed by a medical professional or a positive PCR test for SARS-CoV-2;” Why the antigen test and PCR test in recovering patients with COVID-19 were positive? The authors might have to provide details of the time interval between the onset of infection and the onset of symptoms of post-COVID-19 syndrome in the included patients.

As mentioned in the introduction “Due to the lack of a definition of post-COVID-19 syndrome, the WHO issued Delphi guidelines, which define post-COVID-19 syndrome as a condition that occurs in individuals with a history of SARS-CoV-2 infection, usually 3 months after the onset of infection with symptoms that last at least 2 months and cannot be explained by an alternative diagnosis”. However, in the results (p4 line 115) the authors mentioned " The average time between the onset of infection and the onset of symptoms of post-COVID-19 syndrome was 6.11±11.15 weeks.", which seems to be a contradiction.

Minor editing of English language required

Reviewer 2 Report

The manuscript by Hegna et al., entitled ‘Post-COVID-19 syndrome in neurology patients: a single center experience’ reports a retrospective study on the nervous system involvement in post-covid-19 patients. The authors reported retrospectively collected data for 243 patients in the period from May 2021 to June 2022 using the in-hospital medical records system. 

They described a wide list of symptoms, associated with gender, diagnostic tests recommended to patients, and most frequent consultative examination. 

The discussion is flat and does not stimulate an interpretation of these data, by a relative novelty with respect to current literature. Many studies were published during these covid-19 times, and respect post-covid-19 to date we do not have a clear clinical definition.

In addition, the statistical analysis is poor, and clinical/symptoms correlation was not performed. 

All laboratory and imaging data are only described, but they are more critical to establish an effective correlation between covid-19 /post-covid-19 and comorbidities. The only positivity test to Sars-Cov-2 is not sufficient. All these patients were administered the vaccination and a new covid-19 infection is important to establish.

The retrospective period corresponds to the registration of several virus variants, and to date, we know that which neurological symptoms were more frequently associated.

The follow-up of these patients after vaccination is very important because many neurological impairments were reduced.

The study limitations were described, but they reduced the importance of the conclusions.

The literature cited is appropriate and relevant to the study.

I suggest providing readable information about the tests, such as a reader-friendly table and figure, to better guide the reading of the manuscript.

The length of the paper is commensurate with the message, but it could be considered as a pilot study of a retrospective analysis, and a deep and more statistical accuracy can improve the paper's message. 

Reviewer 3 Report

The study aimed to determine the frequency and characteristics of neurological post COVID-19 syndrome and which diagnostic and therapeutic measures were used in the treatment of these patients. Data were collected for 243 patients examined in the period from 11 May 2021 to 22 June 2022. The inclusion criteria were COVID-19 illness and neurological symptoms associated COVID-19. Exclusive criteria were non-neurological symptoms, patients who did not suffer from COVID-19, and symptoms that occurred after vaccination against the SARS-CoV-2 virus. Data for 227 patients with neurological post COVID-19 symptoms were analyzed. Most patients presented with multiple symptoms, most often headache, cognitive impairment, loss of smell, paraesthesia, fatigue, dizziness and insomnia. Patients were most often referred for consultative examinations, neuroradiological imaging and EEG. The therapy was mostly symptomatic. Most patients had no change in symptoms on follow-up visits (53.21%), while positive outcome was found in 44.95% of patients. This study found that neurological post-COVID-19 syndrome is more common in women and generally the most common symptoms are headache and cognitive impairment. The gender distribution of symptoms was clearly visible and should be further investigated. There is a need for longitudinal follow-up studies to better understand the disease dynamic.

The study has several limitations, including the fact that it was conducted at a single center, which may limit the generalizability of the findings. Additionally, the study only included patients with neurological symptoms associated with COVID-19, which may have excluded patients with other types of neurological symptoms. The study also did not include a control group, which makes it difficult to determine whether the observed symptoms were specifically related to COVID-19. Finally, the study did not include a long-term follow-up period, which may have limited the ability to fully understand the disease dynamic.

This article presents compelling insights and novel findings in its field, making it an interesting read for researchers and practitioners alike.

Figure 2 in the article requires careful attention both in terms of its quality and organization. Improving the clarity and resolution of the figure, as well as reorganizing the elements for better comprehension, would enhance its effectiveness in supporting the article's key points.

To enhance the article's comprehensiveness, it is crucial to include additional information about the COVID-19 situation in Croatia and its post-COVID challenges. Specifically, exploring any unique aspects of the Croatian model or the Specific Rijeka Center could provide valuable insights. It would be worthwhile to discuss if the center's approaches and outcomes are representative of the broader country or if there are specific characteristics that distinguish it. This inclusion would provide a more comprehensive understanding of the Croatian context and contribute to the article's overall relevance and applicability.

In addition to the points mentioned earlier, it is important to address the availability of data in the article. Specifically, it is crucial to clarify whether the data used in the study is publicly available or if there are any restrictions or limitations in accessing it. Providing transparency regarding data availability not only strengthens the article's scientific rigor but also enables other researchers to replicate or build upon the study's findings. Including such information would enhance the article's credibility and facilitate further research in the field.

NA

With some minor revisions to address specific points, this article has the potential to contribute significantly to the existing body of knowledge and deserves publication in a reputable journal.

Round 2

Reviewer 1 Report

None

Author Response

Thank you for your valuable review!

Reviewer 2 Report

My suggestions are only personal opinions, and they are referred to current literature on neurological symptoms in Post-Covid-19 patients. The discussion could be entirely rewritten, and there are not only pending specific parts. Descriptive statistics reduce the importance of these data, and a correlative analysis could improve the clinical data. The correlation between covid-19 /post-covid-19 and comorbidities, vaccination, and a potential second covid-19 infection is important to establish the importance of neurological symptoms. In my opinion, analyzed subjects from May 11th, 2021 to June 22nd, 2022 could be vaccinated, if the vaccination was introduced before in your country. In addition, the record relative to the SARS-Covid-19 variant could aid to interpret the appearance of neurological symptoms. The neuropsychological evaluation by test administration could be resumed in a table. In conclusion, the corrected version of the manuscript does not improve the novelty of the paper.

Author Response

We understand and thank you for your comments. At this time we stand by the discussion as is, and we agree that there are possibilities for improvement in future studies. Unfortunately, we do not have access to all the data regarding the patients, specifically regarding vaccination and variant type, which we have highlighted in the limitations of the paper.